# Multilevel Structure Extraction-Based Multi-Sensor Data Fusion

**Puhong Duan [1] , Xudong Kang [1,*] , Pedram Ghamisi [2] and Yu Liu [3]**

[1]  College of Electrical and Information Engineering, Hunan University, Changsha 418002, China; puhong_duan@hnu.edu.cn
[2]  Helmholtz-Zentrum Dresden-Rossendorf (HZDR), Helmholtz Institute Freiberg for Resource Technology, 09599 Freiberg, Germany; p.ghamisi@hzdr.de
[3]  School of Instrument Science and Opto-electronics Engineering, Hefei University of Technology, Hefei 230009, China; yuliu@hfut.edu.cn
*  Correspondence: xudong_kang@hnu.edu.cn; Tel.: +86-731-8882-2866

**Abstract:** Multi-sensor data on the same area provide complementary information, which is helpful for improving the discrimination capability of classifiers. In this work, a novel multilevel structure extraction method is proposed to fuse multi-sensor data. This method is comprised of three steps: First, multilevel structure extraction is constructed by cascading morphological profiles and structure features, and is utilized to extract spatial information from multiple original images. Then, a low-rank model is adopted to integrate the extracted spatial information. Finally, a spectral classifier is employed to calculate class probabilities, and a maximum posteriori estimation model is used to decide the final labels. Experiments tested on three datasets including rural and urban scenes validate that the proposed approach can produce promising performance with regard to both subjective and objective qualities.

**Keywords:** multi-sensor fusion; hyperspectral image (HSI); multilevel structure extraction; light detection and ranging (LiDAR); synthetic aperture radar (SAR)

## 1. Introduction

With the advance of imaging techniques, the amount of remote sensing data collected by various remote sensors is growing, which allows us to combine multiple types of data for earth observation [1–3]. Such data record different reflectance characteristics, e.g., rich spectral information, high spatial resolution, and height information. The availability of such datasets makes it possible to merge multi-sensor data to further boost the identification accuracy of different materials. For example, a hyperspectral image (HSI) can provide abundant spectral information which is helpful for distinguishing different types of land covers [4–6]. Nevertheless, when the land covers are composed of the same material, i.e., roads and roofs, the spectral curves of such objects are very similar. In this situation, it is hard to distinguish these objects with the same spectral curves using HSI data. Different from the HSI, a light detection and ranging (LiDAR) image can characterize the height and structure information of various objects. Therefore, by integrating the advantages of the two types of data, the fusion of HSI and LiDAR has exhibited better identification performance over single sensor [7–10].

In the past decades, a diversity of multi-sensor fusion techniques has been investigated to boost the spatial resolution of HSI so as to obtain higher classification performance, including RGB and HSI, multispectral (MS) and HSI, HSI and panchromatic (PAN). Integration of RGB/PAN and HSI data aims to improve the spatial resolution of HSI with the help of RGB/PAN data. For example, in [11], a directional total variation model was used to fuse the HSI and RGB images. In [12], a component

decomposition-based method was proposed to enhance resolution of HSIs for the first time. Promising results were obtained to identify different kinds of minerals. In [13], a structure tensor-based image enhancement method was used to improve the spatial resolution of HSI. Integration of MS and HSI involves combining the spatial details of MS and spectral resolution of HSI to obtain high spatial and spectral resolution data. For instance, in [14], a spectral embedding technique was used to fuse MS and HSI, in which the manifold and low-rank structures were exploited. In [15], a novel image decomposition scheme was used to integrate the spatial information of MS and spectral information of HSI, in which the original data was decomposed by coupled sparse tensor factorization technique.

Lately, diverse deep learning techniques have been also applied to enhance the resolution of the HSI [16–18]. In [16], a novel deep cross-modal network was proposed to increase the classification accuracies. This is the first time to introduce the cross-modality learning into networks with the application to the classification of remote sensing data, showing significance of milestone. In [17], a pretrained convolutional neural network (CNN) was used to regularize the fusion issue. In [18], a spatial-spectral reconstruction network was applied to merge MS and HSI, where the spatial information and spectral information is injected into the HSI, respectively.

Different from resolution enhancement techniques, fusion of LiDAR and HSI aims to combine the height information of LiDAR and spectral information of HSI. In such a way, the classification performance can be greatly improved. Many different kinds of schemes have been developed to achieve the integration of LiDAR and HSI, which can be roughly divided into two categories: fusion rule [19–21] and feature extraction [22–24]. For instance, a generalized graph-based fusion rule was constructed to achieve fusion of HSI and LiDAR [19]. Rasti et al. applied a dimension reduction technique, e.g., total variation component analysis, to fuse multiple features [20]. Ghamisi et al. used a composite kernel method to merge spatial information obtained from original images [21]. Besides, many advanced feature extraction techniques have been utilized to characterize the spatial and contextual information from source images. For example, in [22], an extinction profile was exploited to acquire the spatial features from source images, and the sparse and low-rank model was used to fuse the extracted spatial features. In [23], different kinds of features, including spectral and spatial information, were considered to model the spatial information of original images. These techniques have demonstrated that the characterization of spatial features for multi-sensor data is an efficient way to enhance the classification accuracy. Nevertheless, how to improve the discrimination of different objects is still an open problem.

Furthermore, in recent years, deep learning-based schemes have also been applied for fusion of HSI and LiDAR [7,25,26]. For example, a coupled residual CNN was proposed for fusion LiDAR and HSI [25]. In [26], a dual-tunnel CNN is exploited to extract spectral-spatial features, in which a pixelwise affinity architecture was applied to build the correlation of different objects with different elevation information from LiDAR. In [7], deep CNN was employed to classify the fused features obtained by extinction profiles. However, the classification performance of these deep learning-based methods relies heavily on the amount of the labeled training samples.

In this work, a novel fusion approach based on multilevel structure extraction (MSE) is developed, which is comprised of three steps. First, the multilevel structure extraction method is constructed to capture the discriminative spatial features from multiple source images. Then, the low rank representation is adopted to fuse the high-dimensional features into low-dimensional subspace. Finally, the fused features are fed into the MLR classifier to obtain class probabilities, and the maximum posteriori estimation model is used to optimize the class probabilities to yield the final map. Experiments performed on three different scenes validate that the proposed method shows superiority compared to several state-of-the-art fusion approaches. The main contributions of this work is summarized as follows.

- A multilevel structure feature extractor is constructed and exploited to model the spatial and contextual information from input images, which can better characterize the discrimination between different land covers.

- A general multi-sensor fusion framework is proposed based on feature extraction and probability optimization, which can effectively fuse multi-sensor remote sensing data, such as HSI, LiDAR, and synthetic aperture radar (SAR).
- Classification quality of the proposed method is examined on three datasets, which indicates that our method obtains outstanding performance over other state-of-the-art multi-sensor fusion techniques with regard to both classification accuracies and maps. We will also make the codes freely available on author's Github repository: https://github.com/PuhongDuan.

The remaining section of this article is given as follows. The methodology is introduced in Section 2. Section 3 mainly describes the experimental results and analyses. Finally, Section 5 summarizes several concluding remarks.

## 2. Methodology

Figure 1 displays the flow-process diagram of our method, which is comprised of three key steps: First, the multilevel structure extraction is built and is employed to extract the spatial and semantic information from original data. Then, the extracted features are merged with a low rank model. Finally, the MLR classifier is tested on the fused features to obtain class probabilities followed by maximum a posteriori estimation model.

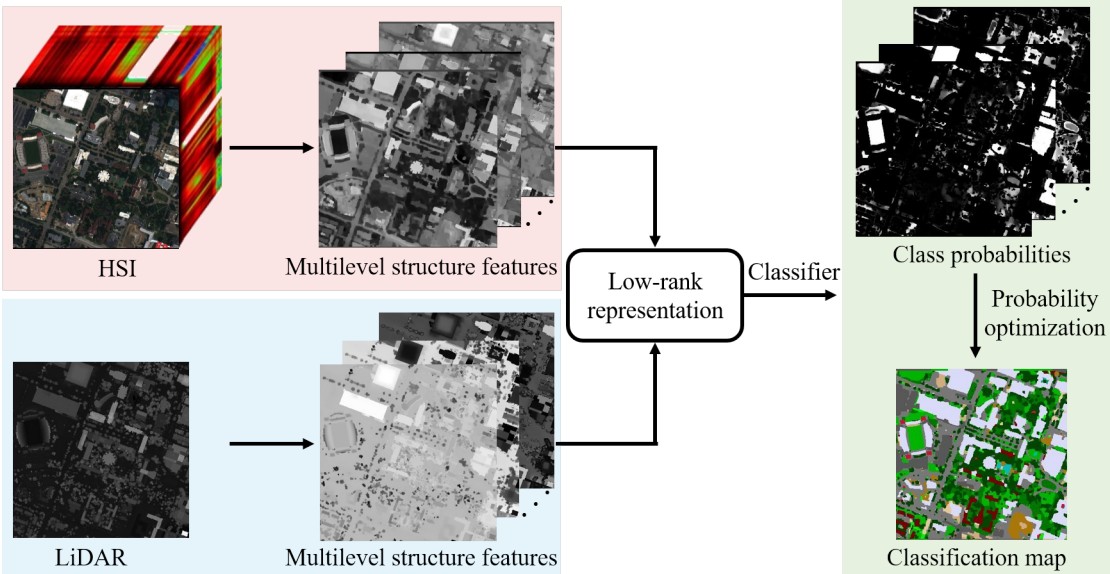

**Figure 1.** The schematic of the proposed multilevel structure extraction method.

### 2.1. Multilevel Structure Extraction

Structure extraction aims at decomposing the input data into two components, i.e., a structure component that reflects the salient spatial structures and a texture component that contains noise and details. However, a single structure feature cannot effectively characterize the spatial characteristics of different kinds of land covers in a scene as different objects usually have different shapes and sizes. To provide a more complete characterization of different objects, a multilevel structure feature extraction approach is developed to characterize the spatial features from multi-sensor data. Specifically, morphological attribute filtering is first used to extract the multilevel architecture of the input because the morphological attribute filtering has been proven to be a multilevel shape-size descriptor [27,28]. Suppose the input $\mathbf{I}$, the spectral dimension of $\mathbf{I}$, is first decreased using principal component analysis (PCA) so as to preserve the first principal component $\hat{\mathbf{I}}$. Then, the multilevel attribute profiles of the dimension reduced image $\hat{\mathbf{I}}$ can be computed as follows,

$$\mathbf{A}(\hat{\mathbf{I}}) = \{\phi^{\lambda_L}(\hat{\mathbf{I}}), \phi^{\lambda_{L-1}}(\hat{\mathbf{I}}), ..., \phi^{\lambda_1}(\hat{\mathbf{I}}), \gamma^{\lambda_1}(\hat{\mathbf{I}}), ..., \gamma^{\lambda_{L-1}}(\hat{\mathbf{I}}), \gamma^{\lambda_L}(\hat{\mathbf{I}})\}, \tag{1}$$

where $\phi$ and $\lambda$ stand for attribute thinning and thickening operators, respectively. $\Lambda = \{\lambda_i | i = 1, 2, ..., L\}$ denotes a series of threshold values for the predefined predicate $T$.

Next, the structure extraction is performed on the multilevel attribute profiles $A(\hat{\mathbf{I}})$ to obtain the multilevel structure features. Specifically, a relative total variation-based structure extraction approach [29] is adopted to remove the texture component from $\mathbf{A}(\hat{\mathbf{I}})$.

$$\arg \min_{\mathbf{S}} \sum_{i=1}^{T} (\mathbf{S}_i - \mathbf{A}_i)^2 + \alpha \cdot \left( \frac{\mathscr{D}_x(i)}{\mathscr{L}_x(i) + \varepsilon} + \frac{\mathscr{D}_y(i)}{\mathscr{L}_y(i) + \varepsilon} \right), \tag{2}$$

where $T$ denotes the amount of image pixels. $\mathbf{S}$ indicates the desired result. $\alpha$ represents a weight which controls the smoothness. $\varepsilon$ is used to avoid dividing by zero. The solution of the energy function in Equation (2) can be found in [29].

Here, $\mathscr{D}_x$ and $\mathscr{D}_y$ denote the total variations (TV) in the horizontal and vertical axes, respectively. This TV mainly measures the spatial differences within the local region $R(i)$.

$$\mathscr{D}_x(i) = \sum_{j \in R(i)} g_{i,j} \cdot |(\partial_x \mathbf{S})_j|$$

$$\mathscr{D}_y(i) = \sum_{j \in R(i)} g_{i,j} \cdot |(\partial_y \mathbf{S})_j| \tag{3}$$

where $\partial_x \mathbf{S}$ and $\partial_y \mathbf{S}$ mean the partial derivatives in the horizontal and vertical axes, respectively. $g_{i,j}$ indicates a weight as follows,

$$g_{i,j} = \exp \left( -\frac{(x_i - x_j)^2 + (y_i - y_j)^2}{2\sigma^2} \right). \tag{4}$$

where $\sigma$ represents the window size. Finally, in order to decrease the computational burden and further improve the discrimination among different land covers, the kernel PCA technique is used to reduce the dimension of the multilevel structure features so as to obtain dimension reduced features, where 80% principal components are preserved for the following fusion in this work.

*2.2. Feature Fusion*

Assume $\mathbf{S}_i, i = 1, 2$ to be the spatial features extracted from source images, a low rank representation method [30] is adopted to merge the extracted spatial information from multi-source images to reduce the redundant information between the extracted features. Accordingly, the low rank model is expressed as

$$\bar{\mathbf{S}} = \mathbf{F} \mathbf{P}^T + \mathbf{N} \tag{5}$$

where $\bar{\mathbf{S}} = [\mathbf{S}_1, \mathbf{S}_2]$ denotes the stacked spatial features. $\mathbf{F}$ is the low-dimensional features, and $\mathbf{P}$ represents the subspace basis. $\mathbf{N}$ stands for the error term. Our goal is to estimate the low-dimensional features $\mathbf{F}$ and subspace basis $\mathbf{P}$ with only $\bar{\mathbf{S}}$. To solve this problem, a total variation prior is adopted to retain the spatial structure of the input. The corresponding energy function is as follows.

$$\arg \min_{\mathbf{F}, \mathbf{P}} \frac{1}{2} \left\| \mathbf{S} - \mathbf{F} \mathbf{P}^T \right\|_F^2 + \beta \|\mathbf{F}\|_{TV} \quad s.t. \ \mathbf{P}^T \mathbf{P} = \mathbf{I} \tag{6}$$

To solve the nonconvex problem Equation (6) mentioned above, a cyclic descent algorithm [31] is adopted, which is as follows.

(1) $\mathbf{F}$ step: By fixing variable $\mathbf{P}$, Equation (6) can be solved by split Bregman algorithm [32]:

$$\mathbf{F}^{m+1} = SplitBregman(\mathbf{G}, \beta). \tag{7}$$

Here, $\mathbf{G} = [g_{(i)}] = \mathbf{S} \mathbf{P}^m$.

(2) **P** step: By fixing variable **F**, Equation (6) becomes an orthogonal Procrustes issue, which can be solved by low-rank Procrustes rotation.

$$\mathbf{P}^{m+1} = \mathbf{Q}\mathbf{R}^T, \tag{8}$$

where **Q** and **R** are calculated by singular value decomposition $\mathbf{S}^T\mathbf{F} = \mathbf{Q}\sum\mathbf{R}^T$.

*2.3. Probability Optimization*

When the fused features **F** are obtained, a spectral classifier, i.e., multinomial logistic regression (MLR), is performed on **F** to obtain the class probabilities $P(Y|\mathbf{F})$. Then, maximum a posteriori estimation is exploited to optimize the class probabilities $P(Y|\mathbf{F})$, which can be calculated as

$$\arg\max P(Y|\mathbf{F}) = \arg\max \prod_i \frac{P(y_i|\mathbf{F}_i)}{P(y_i)} P(Y), \tag{9}$$

where $P(y_i)$ is assumed to be equally distributed. By using the logarithm operation, Equation (9) can be written as

$$\hat{Y} = \arg\max\{\sum_i \log P(y_i|\mathbf{F}_i) + \log P(Y)\}, \tag{10}$$

where $\hat{Y}$ is the final map. $P(y_i|\mathbf{F}_i)$ is the class probabilities obtained by MLR classifier. $P(Y)$ is prior probability which is usually estimated with an isotropic pairwise multilevel logistic model [33].

## 3. Experiments

In the experiment section, three datasets located in rural and urban regions are used to examine the fusion effect of the proposed approach. To illustrate the advantage of the proposed MSE, several state-of-the-art multi-sensor fusion schemes are adopted for comparison, including the MLR classifier [34], orthogonal total variation component analysis (OTVCA) [22], sparse and smooth low-rank analysis (SSLRA) [35], and subspace-based fusion method (SubFus) [36]. For the competitive approaches, we have used exactly similar hyperparameters suggested in their original papers.

(1) MLR [34]: The MLR classifier is a spectral classification method based on logistic regression.

(2) OTVCA [22]: The OTVCA method is a low-rank-based dimension reduction model. First, morphology filters are utilized to extract spatial information from original images. Then, the low-rank-based dimension reduction model is used to fuse the spatial information. Finally, the fused features are fed into a random forest classifier.

(3) SSLRA [35]: The SSLRA method is a feature fusion technique. First, the spatial features of original data are extracted with morphology filters. Then, a sparse and smooth low-rank model is used to fuse the spatial features. Finally, random forest is used as a spectral classifier.

(4) SubFus [36]: The SubFus method is a recently proposed fusion method. Morphology filters are used to extract the spatial information from source images. Then, the spatial features are merged into a low-dimensional subspace. Finally, the fused features are fed into the random forest classifier to obtain the final map.

Furthermore, in order to quantitatively appraise the classification quality of all considered schemes, three widely used quality indexes [37–39], i.e., overall accuracy (OA), average accuracy (AA), and Kappa coefficient, are employed, which are shown as follows.

(1) OA: OA calculates the percentages of correctly identified samples.

$$OA = \sum_{i=1}^{C} \mathbf{M}_{ii}/N \tag{11}$$

where **M** is the confusion matrix. $\mathbf{M}_{ii}$ is the amount of the $i$th class which is identified into the $i$th class. $C$ and $N$ denotes the total amount of classes and test set, respectively.

(2) AA: AA measures the average of the percentage of correctly identified samples for each land cover.

$$AA = \frac{\sum\limits_{i=1}^{C} \left( \mathbf{M}_{ii} \Big/ \sum\limits_{j=1}^{C} \mathbf{M}_{ij} \right)}{C} \tag{12}$$

where $\mathbf{M}_{ij}$ stands for the amount of the *i*th object which is identified into the *j*th object.

(3) Kappa coefficient (Kappa): Kappa indicates the percentage of correctly identified samples corrected by the number of agreements.

$$Kappa = \frac{N \sum\limits_{i=1}^{C} \mathbf{M}_{ii} - \sum\limits_{i=1}^{C} \left( \sum\limits_{j=1}^{C} \mathbf{M}_{ij} \sum\limits_{j=1}^{C} \mathbf{M}_{ji} \right)}{N^2 - \sum\limits_{i=1}^{C} \left( \sum\limits_{j=1}^{C} \mathbf{M}_{ij} \sum\limits_{j=1}^{C} \mathbf{M}_{ji} \right)} \tag{13}$$

### 3.1. Datasets

(1) Trento dataset:

The Trento dataset was taken from a rural area in Trento, a southern city in Italy. It includes an HSI and a LiDAR-derived DSM [22]. The HSI was gained by the AISA Eagle sensor, and the LiDAR DSM was produced using first and last point cloud pulses obtained by the Optech ALTM 3100EA sensor. Both sensors are equipped with a spatial resolution of 1 m and a spatial size of 600 × 166 pixels. The HSI has 63 spectral channels varying from 402.89 to 989.09 nm and its spectral resolution is 9.2 nm. This scene contains six different land covers: Apple Tree, Building, Ground, Wood, Vineyard, and Road. Figure 2 shows the original images, training label, testing label, and class name.

(2) Berlin dataset:

The Berlin dataset was acquired over an urban scene in Berlin, which is composed of an HSI and a SAR. The EnMAP benchmark HSI can be downloaded from the website: http://doi.org/10.5880/enmap.2016.002, which has 244 spectral channels varying from 400 to 2500 nm with spatial resolution of 30 m. Its spatial size is of 797 × 220 pixels. Based on the geographic coordinates, the corresponding region of the SAR image is downloaded from the Sentinel-1 satellite. The SAR image has four polarimetric bands with spatial resolution of 13 m, and its spatial size is of 1723 × 476 pixels. Before performing the fusion operation, the HSI is upsampled to the same size as the SAR image using the linear interpolation operation. Figure 3 presents the original images, training samples, testing samples, and class names.

(3) Houston 2013 dataset:

The Houston 2013 dataset was acquired over an urban region of Houston, USA on 23 June 2012, which is comprised of an HSI image and a LiDAR-derived image. This image was released by the 2013 GRSS Data Fusion Contest [40]. Both the datasets are of 349 × 1905 pixels. The HSI was collected by the Compact Airborne Spectrographic Imager, which records 144 spectral bands varying from 380 to 1050 nm with spatial resolution of 2.5 m. The LiDAR was captured by NSF-funded Center for Airborne Laser Mapping. This scene contains 15 classes of interest. Figure 4 shows the original images, groundtruth, and class names. The amount of training and test set is listed in Table 1.

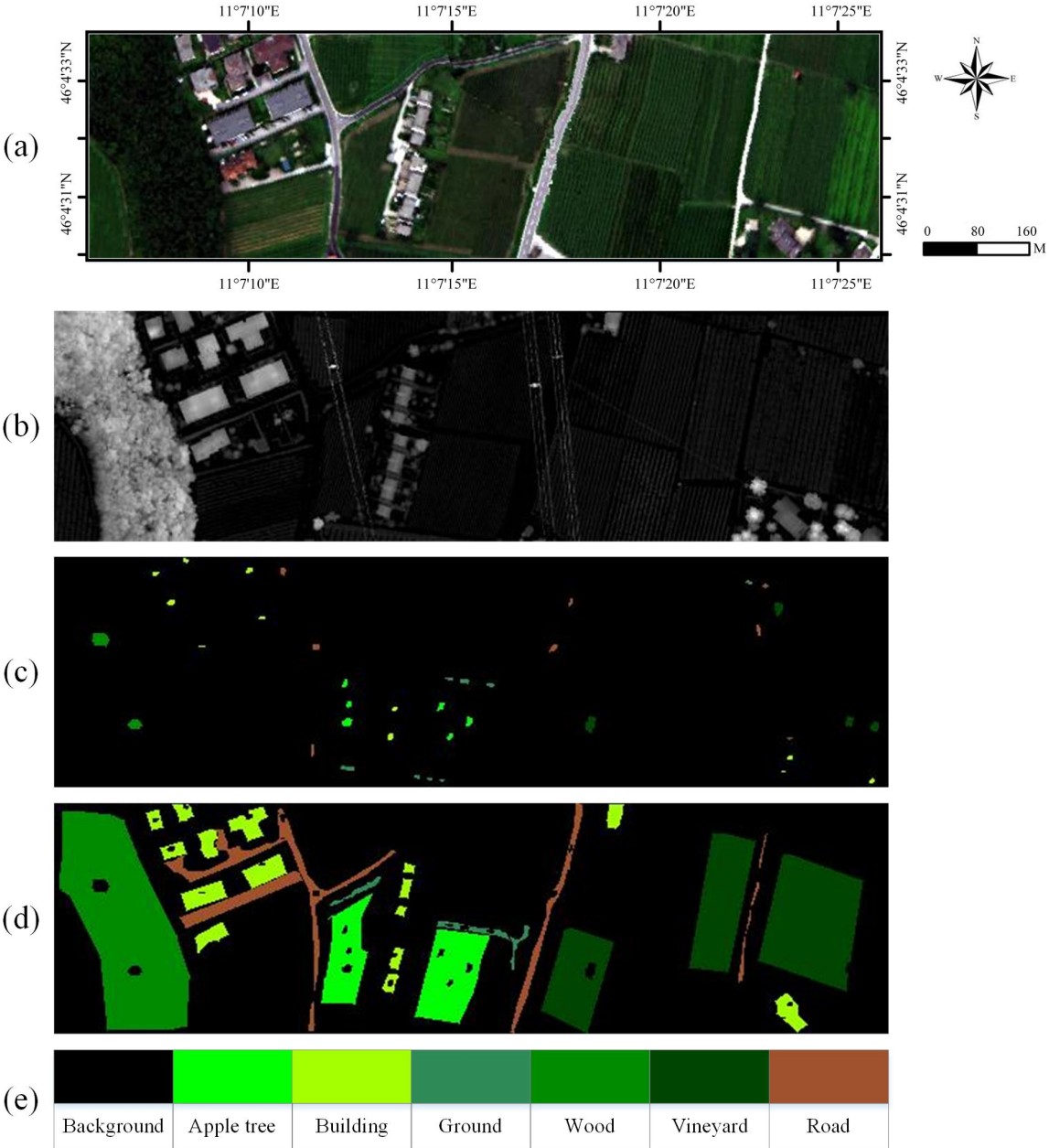

**Figure 2.** Trento dataset. (**a**) False color image of hyperspectral image (HSI) (No. 28, 18, and 8). (**b**) Light Detection and Ranging (LiDAR). (**c**) Training image. (**d**) Testing image. (**e**) Class name.

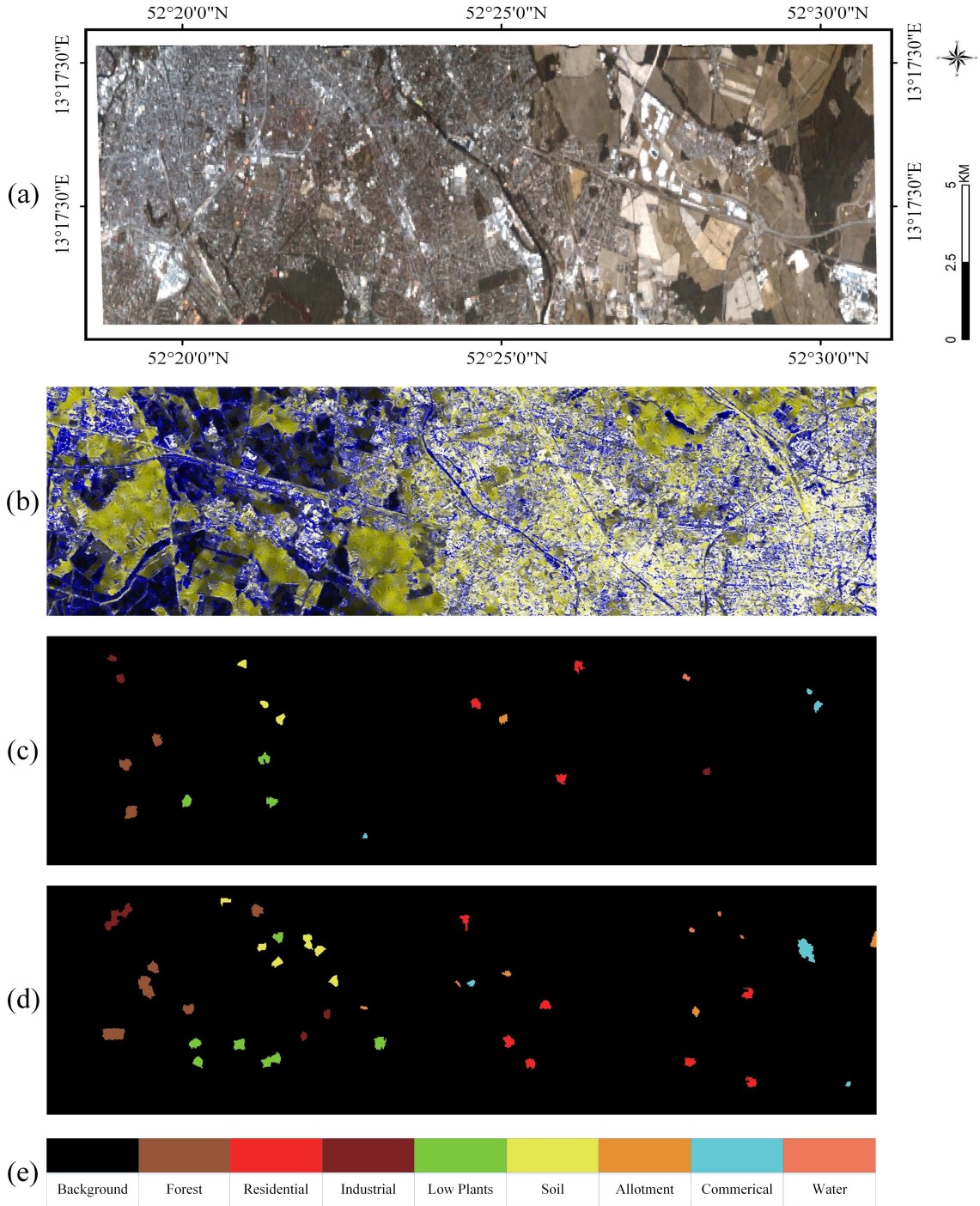

**Figure 3.** Berlin dataset. (**a**) False color image of HSI (No. 30, 20, and 10). (**b**) Synthetic aperture radar (SAR). (**c**) Training image. (**d**) Testing image. (**e**) Class name.

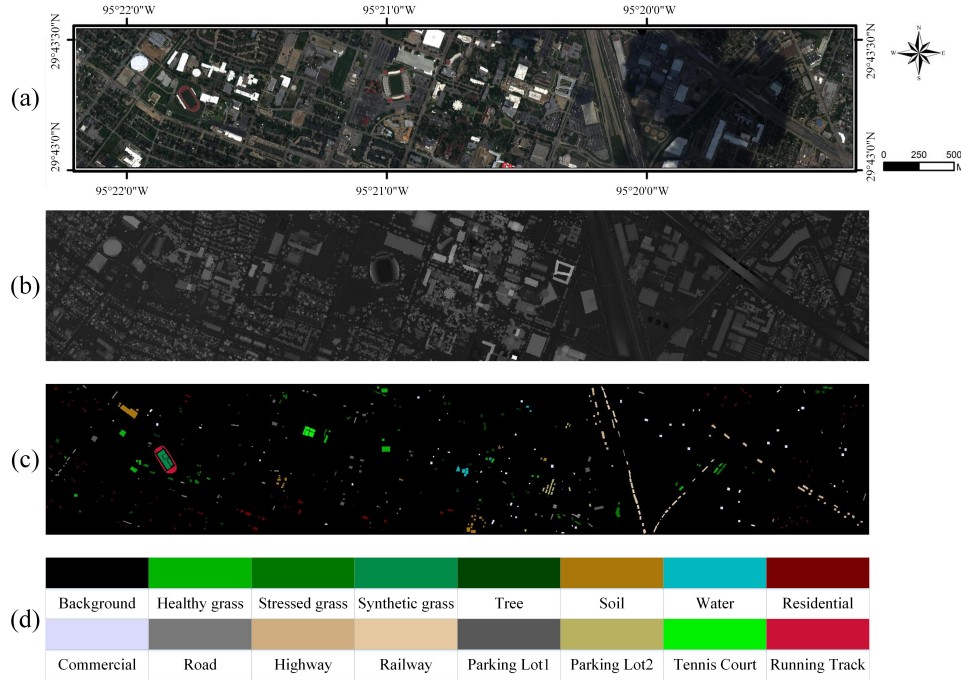

**Figure 4.** Houston 2013 dataset. (**a**) False color image of HSI (No. 60, 40, and 20). (**b**) LiDAR. (**c**) Groundtruth. (**d**) Class name.

**Table 1.** Number of training and test set

| No. | Trento Dataset | | | Berlin Dataset | | | Houston 2013 Dataset | | |
|---|---|---|---|---|---|---|---|---|---|
| | Name | Train | Test | Name | Train | Test | Name | Train | Test |
| 1 | Apple tree | 129 | 3905 | Forest | 1423 | 3249 | Healthy Grass | 50 | 1201 |
| 2 | Building | 125 | 2778 | Residential | 961 | 2373 | Stressed Grass | 50 | 1204 |
| 3 | Ground | 105 | 374 | Industrial | 623 | 1510 | Synthetic Grass | 50 | 647 |
| 4 | Wood | 154 | 8969 | Low Plants | 1098 | 2681 | Tree | 50 | 1194 |
| 5 | Vineyard | 184 | 10,317 | Soil | 728 | 1817 | Soil | 50 | 1192 |
| 6 | Road | 122 | 3252 | Allotment | 260 | 747 | Water | 50 | 275 |
| 7 | Total | 819 | 29,595 | Commerical | 451 | 1313 | Residential | 50 | 1218 |
| 8 | | | | Water | 144 | 256 | Commercial | 50 | 1194 |
| 9 | | | | Total | 5688 | 13,946 | Road | 50 | 1202 |
| 10 | | | | | | | Highway | 50 | 1177 |
| 11 | | | | | | | Railway | 50 | 1185 |
| 12 | | | | | | | Parking Lot1 | 50 | 1183 |
| 13 | | | | | | | Parking Lot2 | 50 | 419 |
| 14 | | | | | | | Tennis Court | 50 | 378 |
| 15 | | | | | | | Running Track | 50 | 610 |
| | | | | | | | Total | 750 | 14,279 |

## 3.2. Classification Results

### 3.2.1. Trento Dataset

Figure 2 exhibits the visual maps obtained by all considered approaches on Trento dataset. The MLR method produces an obvious noisy phenomenon in the classification map because the MLR fails to take the spatial information into consideration (see Figure 2a,b). In addition, the MLR

method on the LiDAR image can better identify the Building and Water classes compared to HSI (Figure 5). This is due to the fact that the height and structure information in the LiDAR image make a great contribution in classifying these land covers. The OTVCA method greatly improves the visual appearance in removing the misclassified noisy labels. Nevertheless, there are still some mislabels in the Vineyard class. The SSLRA method yields an over-smoothed phenomenon for the Road class. The reason is that the SSLRA method performs wavelet transformation before projecting a low-dimensional space. The SubFus method fails to distinguish classes 3 and 5 well, in which many pixels in class 3 are misclassified into class 5. In general, the proposed approach obtains a better visual map in improving homogeneous regions. The key reason is that the constructed feature extractor can better extract the spatial information from HSI and LiDAR data compared to other techniques.

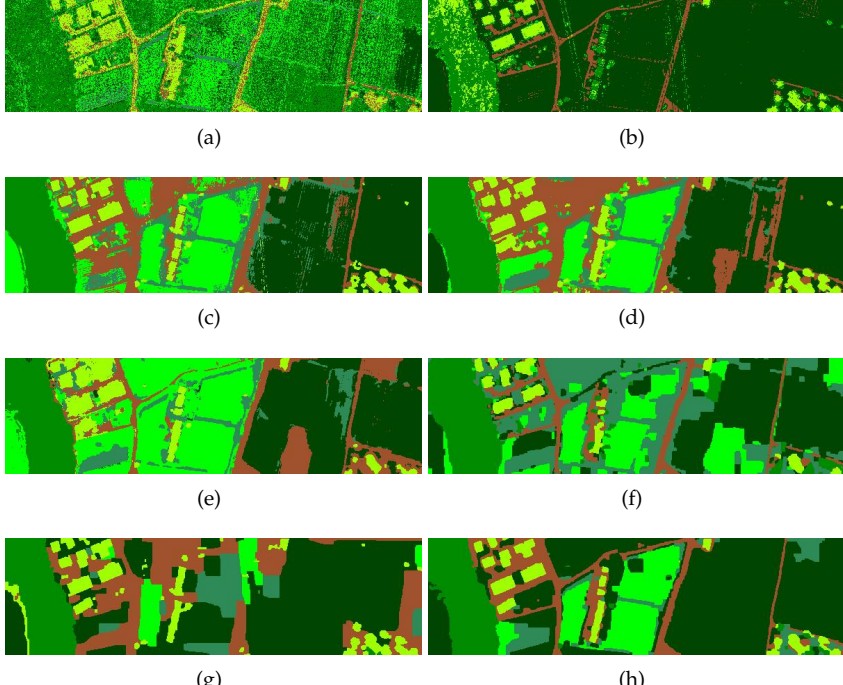

**Figure 5.** Classification maps obtained by different approaches on Trento dataset. (**a**) MLR on HSI. (**b**) MLR on LiDAR. (**c**) OTVCA. (**d**) SSLRA. (**e**) SubFus. (**f**) Our method on HSI. (**g**) Our method on LiDAR. (**h**) Our method.

**Table 2.** Classification accuracies of all considered techniques on Trento dataset.

| Class | MLR | | OTVCA | SSLRA | SubFus | Our Method | | |
| --- | --- | --- | --- | --- | --- | --- | --- | --- |
| | HSI | LiDAR | HSI + LiDAR | HSI + LiDAR | HSI + LiDAR | HSI | LiDAR | HSI + LiDAR |
| Apple tree | 48.53 | 13.97 | 99.60 | 99.88 | **100.0** | 99.36 | 90.29 | 99.60 |
| Building | 58.56 | 53.99 | 95.15 | 95.55 | 97.76 | **97.77** | 93.21 | 97.00 |
| Ground | 84.10 | 0.00 | 43.35 | 80.96 | 98.75 | 54.99 | 43.23 | **100.0** |
| Wood | 59.49 | 90.93 | 99.98 | **100.0** | 99.85 | 99.92 | 99.98 | **100.0** |
| Vineyard | 62.20 | 65.41 | **100.0** | **100.0** | 89.12 | 96.04 | 78.91 | 99.52 |
| Road | 74.03 | 97.15 | 80.68 | 71.94 | 79.55 | 96.69 | 77.68 | **98.27** |
| OA | 59.15 | 72.55 | 95.03 | 95.25 | 93.79 | 96.68 | 86.50 | **99.31** |
| AA | 64.49 | 53.57 | 86.46 | 91.38 | 94.17 | 90.80 | 80.55 | **99.07** |
| Kappa | 46.59 | 61.70 | 93.47 | 93.75 | 91.85 | 95.56 | 81.66 | **99.07** |

Furthermore, the objective quality of different approaches is given in Table 2. It can be observed that the proposed MSE produces the highest objective indexes concerning OA, AA, and Kappa,

which confirms the effectiveness of the proposed MSE from the quantitative perspective. In addition, our method also yields the highest CA in three land covers: Ground, Wood, and Road.

### 3.2.2. Berlin Dataset

Figure 6 presents the visual maps of different techniques on the Berlin dataset. As shown in this figure, the MLR method on the original HSI image still yields "noisy" results (see Figure 6a). The MLR method cannot identify different land covers using the SAR image well (see Figure 6b). This is because the SAR image does not contain rich spectral information. By comparing Figure 6a,f (or Figure 6b,g), it is found that our method on HSI or SAR can effectively improve the classification quality. The reason is that the proposed feature extractor can effectively improve the discrimination between different objects. By fusing the spatial features of HSI and SAR, the proposed approach can produce better classification performance over the single sensor image, e.g., HSI or SAR.

Besides, the objective qualities of different approaches are listed in Table 3. The OTVCA method fails to effectively classify the Water class. The SSLRA and SubFus methods obtain low accuracy on the Allotment class. It is easy to observe that the proposed MSE obtains outstanding classification performance with respect to other techniques. Consequently, the proposed MSE can also be applied to fuse HSI and SAR data.

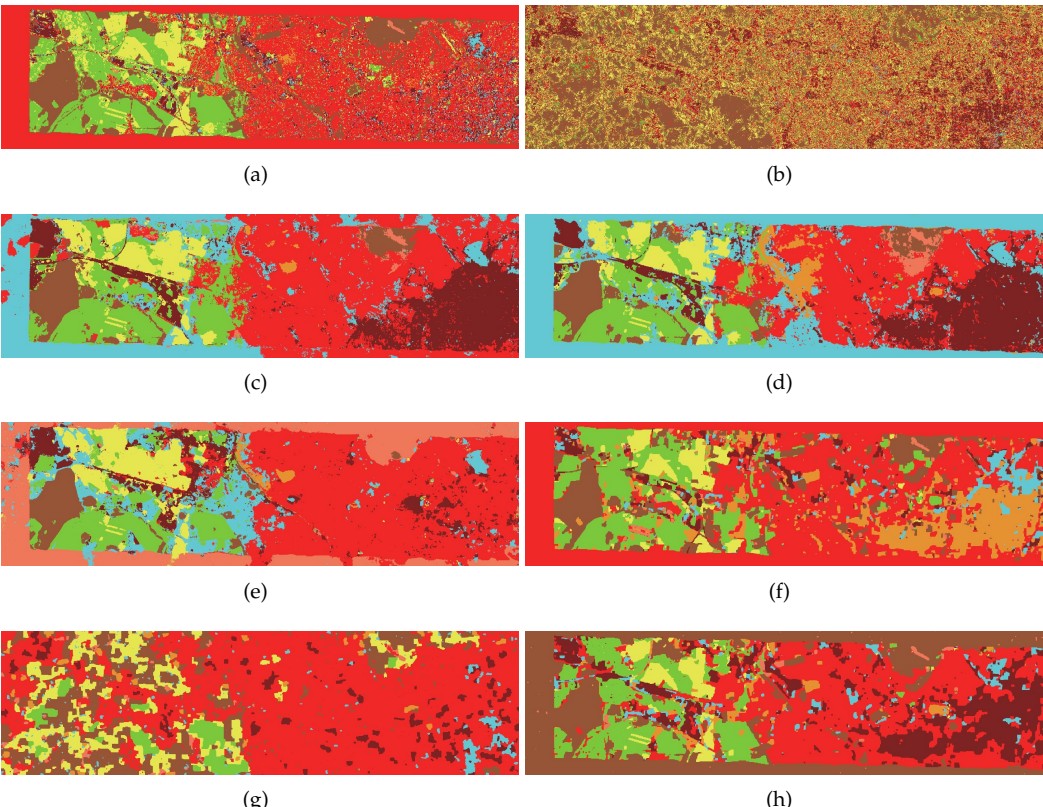

**Figure 6.** Visual maps obtained by different approaches on Berlin dataset. (**a**) MLR on HSI. (**b**) MLR on SAR. (**c**) OTVCA. (**d**) SSLRA. (**e**) SubFus. (**f**) Our method on HSI. (**g**) Our method on SAR. (**h**) Our method.

**Table 3.** Classification accuracies of all considered techniques on Berlin dataset.

| Class | MLR | | OTVCA | SSLRA | SubFus | Our Method | | |
|---|---|---|---|---|---|---|---|---|
| | HSI | LiDAR | HSI + LiDAR | HSI + LiDAR | HSI + LiDAR | HSI | LiDAR | HSI + LiDAR |
| Forest | 90.27 | 33.24 | **100.00** | 99.97 | 97.20 | 97.36 | 41.82 | 99.54 |
| Residential | 69.19 | 34.59 | 63.04 | 76.27 | **85.59** | 76.39 | 51.57 | 73.62 |
| Industrial | 71.81 | 52.33 | 70.76 | 63.45 | 65.17 | **75.11** | 61.33 | 66.25 |
| Low Plants | 91.71 | 11.7 | 91.61 | 98.24 | 94.41 | 86.34 | 47.71 | **94.06** |
| Soil | 91.81 | 22.69 | 94.07 | **100.00** | 89.21 | **100.00** | 44.36 | **100.00** |
| Allotment | 65.54 | 0.00 | **100.00** | 59.24 | 17.27 | 14.33 | **100.00** | 94.37 |
| Commerical | 76.53 | 68.52 | **95.28** | 75.92 | 78.90 | 57.28 | 75.18 | 88.40 |
| Water | 55.88 | 22.22 | 56.96 | **100.00** | 75.78 | 88.39 | 51.16 | **100.00** |
| OA | 82.79 | 32.89 | 84.45 | 86.25 | 83.78 | 79.74 | 50.25 | **87.50** |
| AA | 76.59 | 30.66 | 83.96 | 84.14 | 75.44 | 74.40 | 59.14 | **89.53** |
| Kappa | 79.29 | 17.77 | 81.43 | 83.66 | 80.54 | 75.86 | 40.09 | **85.01** |

### 3.2.3. Houston 2013 Dataset

The third experiment is conducted on a challenging dataset, i.e., Houston 2013. This image consists of various urban land covers, and it is corrupted by shadow. Figure 7 displays the classification maps obtained by different approaches. By observing these classification results, several obvious phenomena can be concluded. First, the spectral classifier, i.e., MLR on HSI or LiDAR, yields different levels of "noise" in the classification maps, while other spatial-spectral feature extraction approaches can greatly improve this problem. Second, the SubFus method cannot identify the shadow region well. In addition, the Grass class is misclassified into the Road class. Third, the proposed method provides better visual map among these considered approaches, in which the edges of different types of objects are well aligned with the labeled land covers. The main reason is that the multilevel structure extraction method can effectively increase the discrimination between different land covers. Furthermore, the classification accuracies listed in Table 4 also confirm the superiority of the proposed MSE.

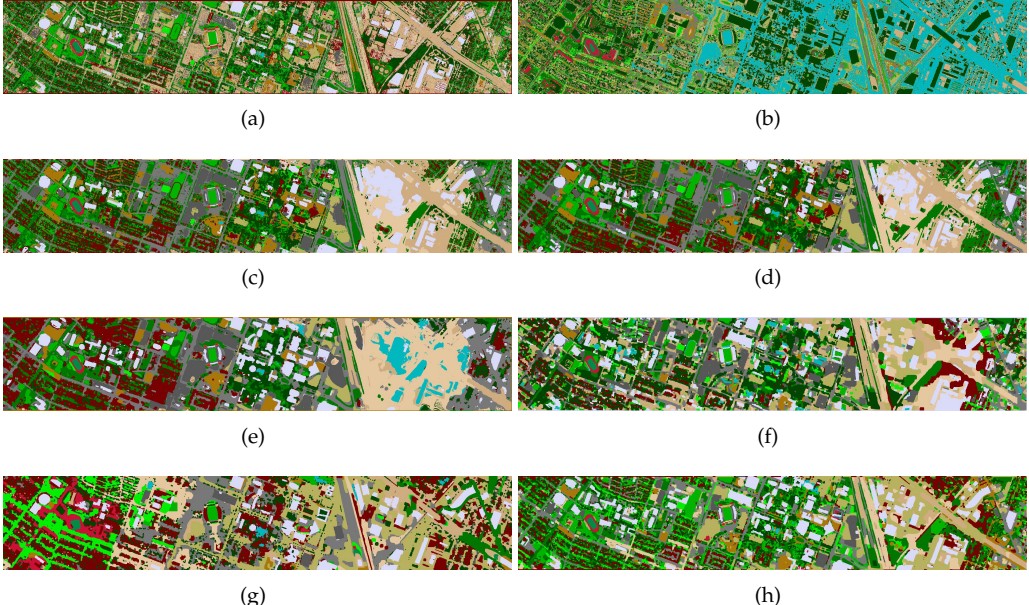

**Figure 7.** Visual maps obtained by different approaches on Houston 2013 dataset. (**a**) MLR on HSI. (**b**) MLR on LiDAR. (**c**) OTVCA. (**d**) SSLRA. (**e**) SubFus. (**f**) Our method on HSI. (**g**) Our method on LiDAR. (**h**) Our method.

**Table 4.** Classification accuracies of all considered techniques on Houston 2013 dataset.

| Class | MLR | | OTVCA | SSLRA | SubFus | Our Method | | |
|---|---|---|---|---|---|---|---|---|
| | HSI | LiDAR | HSI + LiDAR | HSI + LiDAR | HSI + LiDAR | HSI | LiDAR | HSI + LiDAR |
| Healthy Grass | 89.94 | 11.55 | 89.01 | 91.00 | 81.39 | **91.66** | 82.46 | 82.13 |
| Stressed Grass | **98.46** | 8.69 | 89.20 | 86.11 | 80.17 | 80.05 | 2.14 | 94.33 |
| Synthetic Grass | 89.69 | 46.38 | **100.00** | **100.00** | 99.60 | **100.00** | 72.24 | **100.00** |
| Tree | 78.85 | 31.32 | 84.88 | 87.97 | 87.31 | 76.21 | 64.73 | **98.68** |
| Soil | 89.16 | 6.26 | 98.24 | 98.32 | **100.00** | 99.13 | 64.58 | 89.71 |
| Water | **100.00** | 8.61 | **100.00** | **100.00** | **100.00** | 85.60 | 68.84 | **100.00** |
| Residential | 82.78 | 0.00 | 84.44 | 84.23 | 73.32 | 90.45 | 56.43 | **93.69** |
| Commercial | 81.96 | 0.00 | 93.04 | 96.54 | 54.04 | 88.80 | 77.70 | **99.37** |
| Road | 78.09 | 11.40 | 67.08 | 69.70 | 85.74 | 84.73 | 15.70 | **91.78** |
| Highway | 43.66 | 18.16 | 75.30 | 80.29 | 68.53 | 74.64 | 40.00 | **95.71** |
| Railway | 70.50 | 7.43 | 85.05 | 87.30 | **99.72** | 68.84 | 67.60 | 80.25 |
| Parking Lot1 | 85.39 | 0.00 | 89.19 | 87.17 | 74.54 | 88.61 | 87.30 | **96.59** |
| Parking Lot2 | 41.00 | 6.25 | 66.72 | 59.94 | **69.82** | 50.69 | 21.58 | 38.85 |
| Tennis Court | 94.13 | 36.63 | 92.27 | **100.00** | 99.60 | 66.99 | 20.30 | 94.57 |
| Running Track | 93.31 | 40.04 | 99.24 | 97.01 | **100.00** | **100.00** | 58.92 | 97.01 |
| OA | 76.33 | 20.24 | 86.01 | 86.77 | 82.41 | 82.56 | 46.04 | **88.96** |
| AA | 81.13 | 15.52 | 87.58 | 88.37 | 84.92 | 83.09 | 53.37 | **90.18** |
| Kappa | 74.42 | 15.13 | 84.88 | 85.71 | 80.98 | 81.19 | 42.37 | **88.09** |

## 4. Discussion

### 4.1. The Influence of Different Parameters

In this subsection, the influence of different parameters to the performance of the proposed MSE is discussed. In our method, there are three free parameters that need to be determined, including the smoothing parameter $\alpha$, the window size $\sigma$, and the number of the fused features $K$. An experiment is tested on the Trento dataset with standard training and test samples. When $\alpha$ is analyzed, $\sigma$ and $K$ are set to be 2 and 30, respectively. Figure 8a gives the influence of the proposed method with different $\alpha$ to the classification accuracy. We can observe that the objective quality first increases, and then tends to decrease. This is mainly because the smoothing parameter $\alpha$ easily removes some important structural features when $\alpha$ is relatively large. Thus, $\alpha = 0.003$ is selected as the default parameter. When $\sigma$ is analyzed, $\alpha$ and $K$ are fixed, i.e., $\alpha = 0.003$ and $K = 30$. Figure 8b presents the change trend. It is easy to see that the proposed MSE is able to yield promising accuracy when $\sigma = 2$. Similarly, when we discuss the effect of the proposed MSE with different $K$, $\sigma$ and $\alpha$ are fixed. Figure 8c exhibits the influence of the proposed method with different $K$. When $K$ is 30, the proposed MSE yields higher classification accuracy. Therefore, $\alpha$, $\sigma$, and $K$ are set to be 0.003, 2, and 30, respectively, which are regarded as the default parameters for all used datasets in this work.

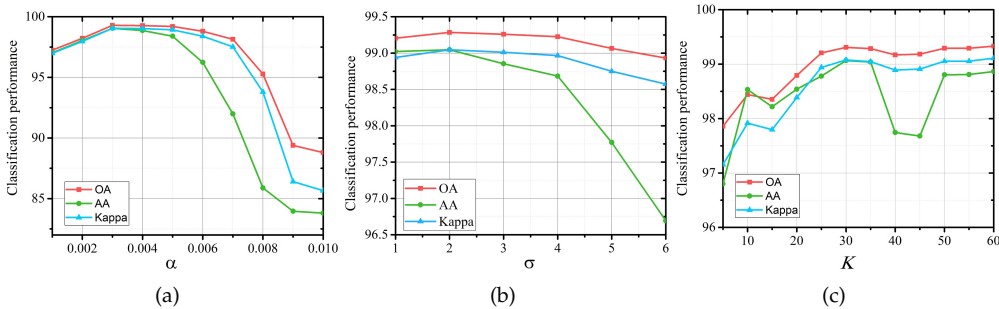

**Figure 8.** The influence of the proposed multilevel structure extraction (MSE) with different parameters. (**a**) The smoothing parameter $\alpha$. (**b**) The window size $\sigma$. (**c**) The number of the fused features $K$.

### 4.2. The Influence of Different Feature Extractors

To prove the superiority of the proposed multilevel structure extraction, several popular feature extraction approaches in remote sensing community are adopted for comparison, including KPCA, structure extraction (SE), intrinsic image decomposition (IID), image fusion and recursive filtering (IFRF), and extended morphological attribute profiles (EMAP). An experiment is tested on the Trento dataset. The classification accuracy of the proposed MSE with different feature extractors is presented in Table 5. It is found that the proposed multilevel structure extractor yields the highest classification accuracies regarding OA, AA, and Kappa coefficient among all considered approaches, which confirms the effectiveness of the proposed feature extractor.

**Table 5.** Classification quality of the proposed MSE with different feature extractors.

|       | KPCA [41] | SE [42] | IID [43] | IFRF [44] | EMAP [45] | MSE       |
|-------|-----------|---------|----------|-----------|-----------|-----------|
| OA    | 97.76     | 98.31   | 97.40    | 95.09     | 97.99     | **99.31** |
| AA    | 94.36     | 97.20   | 95.41    | 84.54     | 97.93     | **99.07** |
| Kappa | 97.03     | 97.74   | 96.52    | 93.52     | 97.31     | **99.07** |

### 4.3. Computing Time

The running time of all considered approaches on three datasets is given in Figure 9. In this work, all experiments are carried out a laptop with 8 GB RAM and 2.6GHz with Matlab 2014a. It is found from Figure 9 that spectral classifier is computationally efficient as it does not have feature extraction and fusion steps. The SSLRA method is relatively time-consuming. This is because the feature fusion step needs to solve a nonconvex objective function. The running time of the proposed approach is moderate. Taking the Trento dataset as an example, the running time is about 316s.

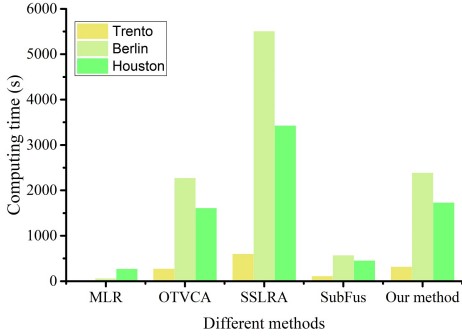

**Figure 9.** Running time of different approaches on three datasets.

## 5. Conclusions

In this study, we developed multilevel structure extraction method for fusion of multi-sensor images. The fusion scheme is composed of three key steps: First, the multilevel structure extraction approach is designed to characterize the spatial and elevation features from the original images. Then, the low rank representation model is exploited to merge the extracted information. Finally, the spectral classifier is conducted on the fused features to produce the class probabilities followed by a maximum posteriori estimation model. Experiments were performed on the rural (Trento) and urban (Berlin and Houston 2013) scenes, which have revealed several conclusions as follows.

(1) The multilevel structure extraction method exhibits outstanding performance in extracting spatial and contextual features from multiple types of data (e.g., HSI, LiDAR, and SAR) compared to other feature extractors.

(2) Experimental results prove that the proposed MSE can considerably boost the classification performance over several popular approaches with regard to both visual maps and objective indexes.

(3) The proposed MSE can yield better classification accuracy compared to single sensor image.

In the future, we will focus on designing novel fusion technique to effectively integrate large-scale multi-sensor data.

**Author Contributions:** P.D. conducted the experiments and wrote the manuscript. X.K. gave some suggestions and revised this work. P.G. carefully improved the presentation of this manuscript. S.L. revised the manuscript. Y.L. modified this manuscript. All authors have read and agreed to the published version of the manuscript.

**Funding:** This research was funded by National Natural Science Foundation of China, grant number 61890962, 61601179, National Natural Science Fund of China for International Cooperation and Exchanges, grant number 61520106001, Natural Science Foundation of Hunan Province, grant number 2019JJ50036, Fund of Key Laboratory of Visual Perception and Artificial Intelligence of Hunan Province, grant number 2018TP1013.

**Acknowledgments:** We would like to thank L. Bruzzone from University of Trento for providing the Trento datset, and thank Danfeng Hong from Technical University of Munich for sharing the Berlin dataset. We acknowledge the support of this manuscript by the National Natural Science Foundation of China (No. 61890962), the National Natural Science Foundation of China (No. 61601179), the National Natural Science Fund of China for International Cooperation and Exchanges (No. 61520106001), the Natural Science Foundation of Hunan Province (No. 2019JJ50036), and the Fund of Key Laboratory of Visual Perception and Artificial Intelligence of Hunan Province (No. 2018TP1013).

**Conflicts of Interest:** The authors declare no conflicts of interest.

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
