# Peer review of "Multilevel Structure Extraction-Based Multi-Sensor Data Fusion"

_remotesensing, doi:10.3390/rs12244034_

Round 1
Reviewer 1 Report
The paper is well written. Some suggestions:
- In the introduction, it is always better to describe in a critic way the state of theart, meaning not only to cescribe past works but also express therir evntual lacks or problems
- In the paragraph where the authors explain the results on the three different dataset, I suggest to explain better the appropriateness of their methd, instead of writing only: "our method on HSI or SAR can effectively improve the classification quality". It is always better to explain why.
- How do they calculate the accuracy? Maybe is better to explain better
- Have th authors the intention to do some other test?
Reviewer 2 Report
General comments:
Puhong Duan and colleagues prosed a novel multilevel structure extraction method to fuse multi-sensor data. The methods includes 3 steps: multilevel structure extraction, spatial information and low-rank model integration, and spectral classification followed by maximum posteriori estimation model. The proposed method (MSE) was then applied on three datasets and showed better performance than other state-of-art multi-sensor fusion methods, including multinomial logistic regression (MLR), orthogonal total variation component analysis (OTVCA), sparse and smooth low-rank analysis (SSLRA), and subspace-based fusion method (SubFus), in terms of overall accuracy, average accuracy, and Kappa coefficient.
Although the findings are interesting and meaningful in remote sensor field, there are still some extra work should be done.
Specific comments:
- Although the methods used for comparison were referenced, authors should briefly describe the key algorithms used in each methods and their strengths and weaknesses of them.
- What are criteria to decide an image as training or test one? Table 1 showed an extreme unbalance in number of each class between the training and test sets. Especially, the numbers of Houston 2013 dataset in this text are smaller than those in other sources 1, 2. This difference may results in lower accuracy compared with other methods such as CResNet-AUX1 and post-classification segmentation 2 when applying in the Houston 2013 dataset.
- It’s clear and mentioned in the text that the performance of methods are affected by the choice of hyperparameters. To make it more reproducible, all hyperparameters in each models/methods (not only MSE) derived during the training steps on each datasets should be reported.
- It would be better if the performance in training sets are also reported along with those in test sets to see how balance the algorithms work.
- Accuracy is sensitive to the cut-off values. How did you choose the cut-off values for each classifier? Are they automatically chosen or manually set up?
- Please provide references for the datasets.
- Please define the evaluation metrics used in the text: OA, AA, and Kappa.
References
- Li, H.; Ghamisi, P.; Rasti, B.; Wu, Z.; Shapiro, A.; Schultz, M.; Zipf, A., A Multi-Sensor Fusion Framework Based on Coupled Residual Convolutional Neural Networks. Remote Sensing 2020, 12 (12), 2067.
- Debes, C.; Merentitis, A.; Heremans, R.; Hahn, J.; Frangiadakis, N.; Kasteren, T. v.; Liao, W.; Bellens, R.; Pižurica, A.; Gautama, S.; Philips, W.; Prasad, S.; Du, Q.; Pacifici, F., Hyperspectral and LiDAR Data Fusion: Outcome of the 2013 GRSS Data Fusion Contest. IEEE Journal of Selected Topics in Applied Earth Observations and Remote Sensing 2014, 7 (6), 2405-2418.
Reviewer 3 Report
17/ This availability of these data
Please rephrase for example , The availability of such datasets allows….
27 In the past several years, a diversity of multi-sensor fusion techniques have been investigated
Well I would suggest , in the past decades ( I remember at least since 1998)
the LiDAR
128 DSM was obtained by the Optech ALTM 3100EA sensor. B
A use of DSM, DEM and an nDSM (normalized difference digital surface model) is not uncommon in Lidar using the information from first and last pulse to separate “elevated vegetation” from “building objects” is almost standard now by the producer. Not to mention full wave (pulse) acquisition
The Lidar is quite incomplete here. And also it’s potential maximum use is not totally exploited.
There is NO scale information on the subsets how many square meters do we discuss in each dataset ?? And be aware
This are snippets of very small subsets of very large datasets.
135 and a SAR [40]. The EnMAP benchmark HSI can be downloaded from the website, which has 244
136 spectral channels varying from 400 to 2500 nm with a GSD of 30 m. Its spatial size is of 797_220 pixels. (with what spatial resolution? please make a scalebar inside the illustrations)
SubFus method fails to distinguish well the Wood and Vineyard classes.
Remark, Such distinction is normally made with large temporal series such as from free Sentinel 2 data or using typical line structure of vineyards in an additional edge detection such as Laplace or Sobel average.
There are classic descriptions of landcover methods seen as standards like Corine or http://www.fao.org/land-water/land/land-governance/land-resources-planning-toolbox/category/details/en/c/1036361/
Please refer to some standards here.
202 set to be 2 and 30, respectively. ( A window size normally is 3 or 5 or 7 please crosscheck!!)
The datasets seem to be “convenient” it is a random selection on free available data. In such case you compare your results with other teams , using the same free data. Who are those other teams ???
